# Microsatellite Instability and BAT-26 Marker Expression in a Mexican Prostate Cancer Population with Different Gleason Scores

**DOI:** 10.3390/diseases13070202

**Published:** 2025-06-30

**Authors:** Ana K. Flores-Islas, Manuel A. Rico-Méndez, Marisol Godínez-Rubí, Martha Arisbeth Villanueva-Pérez, Erick Sierra-Díaz, Ana Laura Pereira-Suárez, Saul A. Beltrán-Ontiveros, Perla Y. Gutiérrez-Arzapalo, José M. Moreno-Ortiz, Adrián Ramírez-de-Arellano

**Affiliations:** 1Laboratorio de Investigación en Cáncer e Infecciones, Departamento de Microbiología y Patología, Centro Universitario de Ciencias de la Salud, Universidad de Guadalajara, Guadalajara 44340, Jalisco, Mexico; ana.flores5605@alumnos.udg.mx (A.K.F.-I.); ana.pereira@academicos.udg.mx (A.L.P.-S.); 2Instituto de Genética Humana “Dr. Enrique Corona Rivera”, Departamento de Biología Molecular y Genómica, Centro Universitario de Ciencias de la Salud, Universidad de Guadalajara, Guadalajara 44340, Jalisco, Mexico; manuel.rico8557@alumnos.udg.mx; 3Laboratorio de Patología Diagnóstica e Inmunohistoquímica, Centro de Investigación y Diagnóstico en Patología, Centro Universitario de Ciencias de la Salud, Universidad de Guadalajara, Guadalajara 44340, Jalisco, Mexico; juliana.godinez@academicos.udg.mx; 4Patología y Nefropatología, Centro de Diagnóstico e Investigación, Guadalajara 44600, Jalisco, Mexico; arisnefropato@gmail.com; 5Unidad Médica de Alta Especialidad, Hospital de Especialidades, Centro Médico Nacional de Occidente, Universidad de Guadalajara, Guadalajara 44340, Jalisco, Mexico; erick.sierra1353@academicos.udg.mx; 6Centro de Investigación y Docencia en Ciencias de la Salud, Universidad Autónoma de Sinaloa, Culiacán Rosales 80030, Sinaloa, Mexico; saul.beltran@uas.edu.mx (S.A.B.-O.); perla.gutierrez@uas.edu.mx (P.Y.G.-A.); 7Departamento de Biología Molecular y Genómica, Centro Universitario de Ciencias de la Salud, Universidad de Guadalajara, Guadalajara 44340, Jalisco, Mexico

**Keywords:** prostate cancer, microsatellite instability, Gleason score, personalized medicine, immunotherapy, Mexico

## Abstract

Background/Objectives: Prostate cancer (PCa) is one of the most common cancers in men worldwide. While standard treatments often provide good initial results, many patients eventually develop resistance and experience a more aggressive relapse. Microsatellite instability (MSI) involves variations in the lengths of microsatellite base repeats in cells. Assessing the frequency of MSI is essential, as it may identify candidates for immune checkpoint inhibitors, which have shown promising outcomes. This study focuses on evaluating the MSI frequency in Mexican PCa patients and exploring its potential relationship with tumor aggressiveness. Methods: In this study, 116 formalin-fixed paraffin-embedded tumoral tissue biopsies from Mexican patients with PCa were collected from Hospital Civil de Culiacán and Pathology and Nephropathology, Diagnosis and Research Center, in the period from 2021 to 2024. The Gleason score was assessed, and the MSI was performed by multiplex PCR with a panel of five markers (NR-27, NR-21, NR-24, BAT-25, and BAT-26). High microsatellite instability (MSI-H) was defined as two or more unstable markers, low microsatellite instability (MSI-L) as an unstable marker, and microsatellite stability (MSS) as no unstable marker. Results: We found 19.83% (23/116) MSI PCa patients, of which 21.74% (5/23) were MSI-H, and 78.26% (18/23) were MSI-L. We found a major distribution of MSI-positive cases (50% (11/22)) in Gleason score 9 patients, corresponding to prognostic group 5. In addition, we found most of the instability in the BAT-26 marker in MSI PCa patients (60.87% (14/23)). Conclusions: This study is the first to evaluate the frequency of MSI in PCa within the Mexican population. Among the Mexican patients with MSI-positive PCa, there was a predominant Gleason score 9 and a majority instability of the BAT-26 marker.

## 1. Introduction

Prostate cancer (PCa) remains one of the leading malignancies worldwide; in 2022, it was the second most frequent cancer among men, with approximately 1.4 million new cases and an estimated 397,430 deaths [1,2]. The aggressiveness of PCa is typically assessed using the Gleason score, which evaluates histological patterns depending on the degree of differentiation [3]. Although PCa initially responds well to local treatments and androgen deprivation therapies (ADTs), most tumors eventually progress to an androgen-independent state, known as castration-resistant prostate cancer (CRPC). CRPC is a more aggressive form of PCa and is significantly harder to treat with conventional therapies. In this context, immunotherapy offers a promising opportunity to transform a challenging cancer into one that is more manageable by enhancing the immune response against tumor cells. A notable example of immunotherapy is the use of immune checkpoint inhibitors (ICIs), such as programmed death-1 (PD-1)/programmed death-ligand 1 (PD-L1) inhibitors.

Microsatellites, also called short tandem repeats (STRs), are short repeated sequences from one to six nucleotides within both coding and non-coding regions of DNA. These sequences represent about 1 to 3% of all the human genome. Microsatellite instability (MSI), a type of genomic instability, is caused by defects in DNA mismatch repair (MMR) genes, in which the length of MS is modified [4,5]. Due to its association with genomic instability, leading to increased mutation rates and tumor progression, MSI has been extensively studied in various cancers, particularly colorectal cancer, where it serves as a key biomarker.

For MSI determination, various techniques are employed, including immunohistochemistry (IHC), PCR panels with different markers of mono-, di-, and even penta-nucleotides, as well as next-generation sequencing (NGS). The gold standard is a fluorescent PCR pentaplex panel, followed by fragment length analysis by capillary electrophoresis, which detects the variability of length of the tested microsatellite markers [6]. By global standards recommendations, two or more unstable markers were defined as microsatellite instability-high (MSI-H), an unstable marker was determined as microsatellite instability-low (MSI-L), and no unstable marker as microsatellite stability (MSS) [7].

Although the role of microsatellite instability (MSI) in PCa is less defined compared to other cancers, its presence, though low (2.6–13.63%) [8,9,10,11,12,13,14,15], offers valuable insights into tumor biology and potential therapeutic strategies. For comparison, MSI prevalence is notably higher in other cancers, such as endometrial (20–31.37%) [16,17,18,19], gastric (8.3–22%) [16,17,18,20,21], and colorectal cancer (15–20%) [16,17,18,22]. MSI-positive tumors are particularly important to identify due to their association with favorable responses to immune checkpoint inhibitors, which underscores their relevance in immunotherapy.

Beyond the variability in MSI prevalence across cancer types, it is crucial to consider differences among populations. Most MSI research in PCa has focused on non-Hispanic white populations, leaving a gap in understanding MSI prevalence and its clinical implications in other ethnic groups. This gap is particularly relevant in Mexican populations, where genetic and environmental factors may influence both the frequency and significance of MSI in PCa, highlighting the need for population-specific studies.

A report by Angel et al. (2024) explored the genomic landscape of metastatic prostate cancer (mPC) in 349 Latin American patients, including 9 Mexican patients (2.6%). Of the 22 patients tested for MSI, 3 (13.63%) were found to have high MSI (MSI-H) [15]. This study represents one of the most significant steps toward understanding the prevalence and implications of MSI in Mexican PCa patients, though more research is needed to expand on these findings.

The Gleason score is a well-established predictor of tumor aggressiveness in PCa. However, integrating molecular markers, such as MSI, can enhance prognostic accuracy and inform personalized treatment strategies. This study investigates the prevalence of MSI in PCa patients within a Mexican population, emphasizing the importance of MSI detection for predicting tumor aggressiveness and guiding treatment decisions.

The correlation between MSI and Gleason scores is of particular interest, as it holds the potential to deepen our understanding of the biological mechanisms underlying PCa and to refine personalized care. Detecting MSI not only aids in identifying aggressive tumors but also provides critical information for selecting appropriate therapeutic options. MSI-positive tumors are strong candidates for immunotherapy, a treatment approach gaining traction in oncology. By elucidating the prevalence and clinical impact of MSI in Mexican PCa patients, researchers aim to optimize therapeutic outcomes and improve patient care through tailored interventions.

## 2. Materials and Methods

### 2.1. Formalin-Fixed Paraffin-Embedded Tumoral Tissue

Formalin-fixed paraffin-embedded tumoral tissue biopsies from Mexican patients with PCa were collected from two sources: (1) Hospital Civil de Culiacán and (2) Pathology and Nephropathology, Diagnosis and Research Center, in the period from 2021 to 2024. The Gleason score for histopathological analysis was assessed by certified pathologists according to ISUP 2019 [3] as a part of the conventional diagnostic.

### 2.2. DNA Extraction

The DNA was extracted using a commercial kit (High Pure PCR Template Preparation Kit, ID: 11 796 828 001) from formalin-fixed, paraffin-embedded blocks tumoral tissue. For DNA extraction, 5-micron sections derived from the paraffin blocks were prepared and subsequently placed into 2 mL tubes. The DNA was quantified by 260/280 nm spectrophotometry (NanoDrop 2000, Thermo Fisher Scientific, Waltham, MA, USA) to ensure a concentration greater than 19.62 ng/mL.

### 2.3. Microsatellite Instability (MSI) Analysis

MSI analysis was performed by multiplex PCR using the Type-it Microsatellite PCR kit (Qiagen ID: 206243) with a panel of 5 markers: NR-27, NR-21, NR-24, BAT-25, and BAT-26. PCR reactions were performed using 100 ng/µL of DNA in a volume of 12 µL. The primer mix was used with a concentration of 100 µM from each primer. PCR conditions included an initial denaturation at 95 °C for 5 min, then 95 °C for 30 s, 58 °C for 90 s, and 72 °C for 30 s for 40 cycles, and a final extension at 60 °C for 30 min. Then, 2 μL of PCR products was taken and mixed with formamide and GeneScan 500 LIZ Dye Size Standard. Samples were heated at 96 °C for 5 min, followed by a thermal shock on ice; after that, samples were analyzed by the SeqStudio Genetic Analyzer (Applied Biosystems, Thermo Fisher Scientific, Waltham, MA, USA).

### 2.4. Microsatellite Instability Interpretation

MSI electropherogram interpretation was performed in the Microsatellite Analysis Thermo Fisher ConnectTM software version 1.2-PRC-build08. Criteria to consider each marker stable were NR-27 (84–90 bp), NR-2 (106–112 bp), NR-24 (128–134 bp), BAT-25 (150–156 bp), and BAT-26 (180–186 bp). Each marker analyzed whose size was smaller or larger than the reference stable marker was considered unstable.

Two or more unstable markers were defined as microsatellite instability-high (MSI-H), an unstable marker was determined as microsatellite instability-low (MSI-L), and no unstable marker as microsatellite stability (MSS) [7].

### 2.5. Statistical Analysis

Statistical analysis was performed using GraphPad Prism version 8.0.1 and R version 4.4.1. The association analysis between (1) the Gleason score groups and microsatellite instability category and (2) the Gleason score groups and microsatellite instability high and low categories was evaluated using Fisher’s exact test extended version and using a Monte Carlo simulation with 100,000 iterations to obtain an exact *p*-value. The differences between each of the markers were analyzed using 2 × 2 tables in OpenEpi (https://www.openepi.com/, accessed on 1 May 2025), and *p*-values < 0.05 were considered statistically significant.

## 3. Results

### 3.1. Gleason Score Distribution in Prostate Cancer Patients

A total of 116 samples were collected, 113 of which had enough available information for Gleason score and prognostic group stratification. Of these 113 patients, the major distribution was Gleason score 9 (*n* = 36, 31.86%), followed by Gleason score 7 (3 + 4) (*n* = 30, 26.55%), then Gleason score 6 (*n* = 22, 19.47%), Gleason score 7 (4 + 3) (*n* = 13, 11.50%), Gleason score 8 (*n* = 10, 8.85%), and the lowest Gleason score represented was 10 (*n* = 2, 1.77%) (Figure 1).

### 3.2. Microsatellite Instability and Their Distribution According to Gleason Score in Mexican Prostate Cancer Patients

Out of 116 cases of PCa patients analyzed, 19.83% (23/116) were positive for MSI, and 80.17% (93/116) were microsatellite stable (MSS) (Figure 2a). Regarding each Gleason score group, we only had Gleason score data from 113 patients: we found that Gleason score 9 exhibited the major ratio of MSI-positive cases, with 11 out of 36 patients, followed by Gleason score 7 (3 + 4) with 6 MSI-positive cases of 30, then Gleason score 6 showed 4 MSI patients of 22, and the last one was Gleason score 8 with 1 of 10 MSI-positive cases. The patients with Gleason scores of 7 (4 + 3) and 10 did not show MSI-positive cases (Figure 2b). According to the MSI distribution among Gleason scores, we found a predominant representation of Gleason score 9, which represented 50% (11/22) of all MSI-positive cases, followed by Gleason score 7 (3 + 4) with 27.77% (6/22), Gleason score 6 with 18.18% (4/22), and Gleason score 8 with 4.55% (1/22) (Figure 2c,d).

The extended version of Fisher’s exact test was used because the contingency table generated between the Gleason score (with six categories: 6, 7 (3 + 4), 7 (4 + 3), 8, 9, and 10) and the MSI/MSS classification (two categories) exceeded the 2 × 2 format. Furthermore, several cells had small or even zero frequencies, which violates the assumptions required to apply the chi-square test of independence (Table 1). Results from the extended Fisher’s exact test showed a *p*-value of 0.2376 (*p* ≥ 0.05), suggesting no statistically significant link between the two variables. The calculated Cramer’s V value was 0.2498, indicating a weak correlation between the Gleason score and microsatellite instability status.

### 3.3. High Microsatellite Instability Presence Related to Gleason Score

Out of 23 cases of MSI PCa patients, 21.74% (5/23) were microsatellite instability-high (MSI-H), and 78.26% (18/23) were microsatellite instability-low (MSI-L) (Figure 3a). Of the 23 MSI patients, we had Gleason score information for only 22 patients. Patients with Gleason score 6 showed 25% MSI-H (1/4) and 75% MSI-L (3/4), Gleason score 7 (3 + 4) patients exhibited 16.67% (1/6) MSI-H and 83.33% (5/6) MSI-L, Gleason score 8 patients did not show MSI-H but 100% presented MSI-L (1/1), and the most representative Gleason score category, 9, displayed 18.18% (2/11) MSI-H-positive cases and 81.82% (9/11) MSI-L-positive cases. Neither Gleason score 7 (4 + 3) nor Gleason score 10 exhibited MSI-positive cases and, therefore, there were no MSI-H/MSI-L cases to report (Figure 3b).

Since the contingency table generated from the Gleason score (with six categories: 6, 7 (3 + 4), 7 (4 + 3), 8, 9, and 10) and the MSI-L/MSI-H classification (two categories) goes beyond the 2 × 2 format, we used the extended version of Fisher’s exact test. Additionally, some cells had very small or zero frequencies, which do not meet the assumptions needed for the chi-square test of independence (Table 2). Results from the extended Fisher’s exact test showed a *p*-value of 1.0 (*p* ≥ 0.05), suggesting no statistically significant link between the two variables. Strength of association measures showed a calculated Cramer’s V value of 0.127, indicating a weak correlation between the Gleason score and microsatellite instability-low (MSI-L) and microsatellite instability-high (MSI-H).

### 3.4. Unstable BAT-26 Marker Higher Among Microsatellite Instability Markers Evaluated

The markers evaluated to determine microsatellite instability were NR-27, NR-21, NR-24, BAT-25, and BAT-26. Of all the evaluated markers, we found a predominant instability in the BAT-26 marker, with 60.87% (14/23) instability among MSI patients. The second unstable marker was NR-21, with 30.43% (7/23) instability. NR-24 and BAT-25 showed the same percentage of instability, 21.74% (5/23). NR-27 did not exhibit instability (Figure 4a). Considering that BAT-26 was the most unstable marker, a comparison was made with the instability frequency observed in the other markers. Statistically significant differences were found between BAT-26 vs. NR-24, NR-25, and NR-27 (*p* < 0.05), but not between BAT-26 vs. NR-21 (*p* = 0.1168). Concerning MSI-H and MSI-L, BAT-26 also had a major percentage of instability in both conditions: it exhibited 80% (4/5) MSI-H and 55.56% (10/18) MSI-L. The second marker was NR-21, with 60% (3/5) instability for MSI-H and 22.22% (4/18) for MSI-L. NR-24 and BAT-25 showed the same results, with 60% (3/5) instability for MSI-H and 11.11% (2/18) for MSI-L. Because NR-27 did not present instability, there are no results to report concerning MSI-H or MSI-L (Figure 4b).

A total of 23 samples were analyzed for both BAT-26 stability and MSI status. The association between BAT-26 stability status and MSI classification was evaluated using Fisher’s exact test, which is suitable for 2 × 2 contingency tables with small sample sizes and low expected frequencies (Table 3). Fisher’s exact test showed no statistically significant association between the BAT-26 stability status and MSI classification (*p* = 0.611). The odds ratio was 3.056 (95% CI: 0.235–176.757), suggesting that while samples with unstable BAT-26 exhibited a numerically higher proportion of MSI-H compared to stable BAT-26 samples, this difference was not statistically significant. Further measures of association confirmed these findings, with Cramer’s V = 0.207 and contingency coefficient = 0.202, both indicating a weak association strength. The Pearson chi-square statistic was 0.982 (df = 1, *p* = 0.322), consistent with the results of Fisher’s exact test.

## 4. Discussion

Prostate cancer (PCa) is one of the most prevalent cancers in men worldwide. Despite conventional treatment having a good initial response, most patients will develop resistance to therapy and relapse with a more aggressive type of cancer, such as castration-resistant PCa. Microsatellite instability (MSI) is a condition in which certain cells show changes in the length of the microsatellite (MS) base repeats. Establishing the frequency of MSI is important because it may make one a candidate for immune checkpoint inhibitors, such as Pembrolizumab (KEYTRUDA, Merck & Co., Inc., Rahway, NJ, USA), a personalized treatment that has been showing promising results.

In this study, we focused on evaluating the MSI frequency of Mexican PCa patients and on the determination of whether there is a relation with tumor aggressivity. We found 19.83% (23/116) MSI PCa patients, which is higher than the range of MSI in PCa reported in the last ten years from 2.6% to 13.63%. These differences could be mainly due to the diagnostic techniques for MSI. The most used is a PCR panel with different markers of mono-, di-, and even penta-nucleotides, immunohistochemistry (IHC), or next-generation sequencing (NGS). Pritchard et al. (2014) [8], using a PCR panel comprised of NR-21, BAT-26, BAT-25, NR-24, and MONO-27, reported 12% (7/60) of patients with mutations in MMR genes and MSI. Abida et al. (2019) [9], using NGS and IHC techniques, reported 3.1% (32/1033) MSI-H/dMMR. In the same year, Chung et al. (2019) [10], using comprehensive genomic profiling (CGP), found 2.6% (87/3326) MSI-H tumor PCa patients. Also, Fraune et al. (2019) [11] reported their results of 3.5% (7/200) dMMR/MSI advanced PCa patients by IHC in a tissue microarray (TMA) format with a confirmation of the suspicion of dMMR by Bethesda PCR panel with BAT-25, BAT-26, D2S123, D5S346, and D17S250 markers. The next year, Zhang et al. (2020) [12], using NGS techniques, found 2.9% (7/241) MSI-H PCa patients. Another study was performed by Hwang et al. (2023) [13]. They separated their metastatic CRPC patients into non-Hispanic black and non-Hispanic white men groups and found a higher dMMR/MSI-H percentage in black men (9.1% (18/198)) over white men (4.9% (36/739)). Lenis et al. (2024) [14] reported 2.8% (63/2257) MSI-H/dMMR PCa detected by NGS techniques. Finally, Angel et al. (2024) [15] found 13.63% (3/22) MSI-H.

This study represents the first to evaluate the MSI frequency in PCa, focusing on the Mexican population. The closest study focused on MSI determination in the Mexican population was performed by Angel et al. (2024). That study included Latin metastatic PCa patients from Argentina (47% (164/349)), Chile (4% (14/349)), Colombia (42.1% (147/349)), Mexico (2.6% (9/349)), and Peru (4.3% (15/349)) [15]. Although Angel et al. included Mexican patients, the details about their MSI determination were not indicated, such as the technique used or the ethnicity of the MSI-positive patients. Furthermore, the Mexican patients were not represented enough and, therefore, the results reported may not reflect the reality of Mexican PCa MSI-positive patients.

We were interested in observing the relationship between microsatellite instability presence and the Gleason score groups, and we found a majority distribution of MSI-positive cases (50% (11/22)) with Gleason score 9, corresponding to Grade Group 5. Our results are consistent with previous reports by Lenis et al. (2024) [14], in which they found MSI-H/dMMR PCa tumors were more common (62%) in Gleason Grade Group 5.

Another focus of this work was determining the distribution of high microsatellite instability (MSI-H) and low microsatellite instability (MSI-L) in PCa patients. Out of all MSI PCa patients, we found 21.74% (5/23) MSI-H, and 78.26% (18/23) were MSI-L. Most of the MSI reports do not consider MSI-L cases and only include MSI-H patients, as in Pritchard’s study, which reported 12% MSI cases without considering MSI-L cases. They were only considered MSI-positive if two or more markers were unstable by PCR panel analyses [8]. If the parameter to determine MSI was MSI sensor score classification—that is, a PCR panel was not used—the studies only considered MSI-positive cases if the score was ≥10, and they reported it as MSI-H, without considering the MSI-L category. Examples of this determination were reported by Abida et al., who reported 3.1% MSI-H/dMMR [9], and Lenis et al., who reported 2.8% MSI-H/dMMR [14]. Other studies did not have enough details about the techniques of MSI determination and neither the distribution of MSI-H nor MSI-L, such as the study of Angel et al. [15].

This study represents the first attempt to evaluate microsatellite instability in the Mexican prostate cancer population, specifically distinguishing between high and low instability. The findings were significant, particularly given that only patients with high microsatellite instability (MSI-H) are considered candidates for immunotherapy, while those with low instability (MSI-L) are not. For instance, The National Comprehensive Cancer Network (NCCN) in their Prostate Cancer Guidelines Version 1.2025 only considers MSI-H (specifically MSI-H, not MSI-L), dMMR, and TMB-high tumors as somatic predictive for immunotherapy with pembrolizumab, and prognostic biomarkers for metastatic castration-resistant prostate cancer (mCRPC), based on the favorable and sustained response [23]. Understanding the prevalence of both MSI-H and MSI-L could enhance our comprehension of prostate cancer’s nature, its association with microsatellite instability, and the role of the immune response in these conditions.

In that context, one emerging market suggesting a link between MSI-L states and an immunologically favorable response, thereby enhancing the clinical relevance of MSI-L tumors, is elevated microsatellite alterations at selected tetranucleotide repeats (EMAST). Recent studies indicate that EMAST can coexist with the MSI-L phenotype and act as a modifying factor, influencing tumor progression and therapeutic response. These studies have identified EMAST as a distinct form of genomic instability, separate from the traditional MSI associated with mismatch repair (MMR) deficiencies. EMAST is characterized by instability at specific tetranucleotide loci and has been observed in various tumor types, including colorectal, lung, and bladder cancers. Its presence is often associated with MSH3 dysfunction and chronic inflammation, and it may influence tumor behavior and responses to immunotherapy. While the clinical significance of EMAST remains under investigation, its potential as a biomarker for prognosis and therapeutic response is garnering increasing attention [24]. In the context of prostate cancer, EMAST has been reported, although infrequently, and its relationship with traditional MSI and MMR status is not fully understood. Given these considerations, future research should explore the prevalence and implications of EMAST in prostate cancer, particularly among specific populations, such as Mexican patients, to enhance our understanding of tumor biology and inform personalized treatment strategies.

Contrary to our findings of 21.74% (5/23) MSI-H and 78.26% (18/23) MSI-L, one of the studies that mention MSI-L-positive patients, by Fraune et al., reported that from the total of seven MSI patients determined by PCR analysis using a Bethesda panel (BAT-25, BAT-26, D2S123, D5S346, and D17S250), they were able to determine the MSI status in six of these: 66.66% (4/6) were MSI-H and 33.33 (2/6) were MSI-L [11]. Another study that reported MSI-L patients was conducted by Chung et al. The authors first reported 2.6% (87/3326) of MSI-H patients, and then as part of other determinations of their own study, they made a group of 82 patients who presented the MMR mutation under LOH or homozygous deletion. In this group, they made a distinction between MSI-H (70% (57/82)), MSI-L (13% (11/82)), and MSS (17% (14/82)) patients [10]. Although the MSI-H and MSI-L percentages reported by Chung et al. were part of a niche result from another determination, the behavior of their results is also contrary to our findings, as these studies showed a predominance of MSI-H over MSI-L. A huge source of microsatellite instability prevalence variety reported is the technique used to determine it and the PCR panel used. There are a variety of PCR panels used to determine microsatellite instability, such as the Bethesda panel [25], which uses the mononucleotides BAT-25 and BAT-26, and the dinucleotides D5S346, D2S123, and D17S250, the variation proposed by Suraweera et al. [26], which considered the quasi-monomorphic mononucleotides BAT-25, BAT-26, NR-21, NR-22, and NR-24, and the variation proposed by Goel et al. [7], conformed by NR-27, NR-21, NR-24, BAT-25, and BAT-26 markers. The markers used in our study to determine MSI were the quasi-monomorphic mononucleotides NR-27, NR-21, NR-24, BAT-25, and BAT-26, proposed by Goel et al. [7]. We found most of the instability in the BAT-26 marker in MSI PCa patients (60.87% (14/23)) and the same behavior in MSI-H and MSI-L PCa patients (MSI-H: 80% (4/5); MSI-L: 55.56% (10/18)). Our results are similar to those reported by Pritchard et al. [8], in which 71.4% (5/7) of MSI human primary and metastatic PCa tissues detected by MSI-PCR (Promega panel: NR-21, BAT-26, BAT-25, NR-24, and MONO-27) were positive for BAT-26.

BAT-26, as the only marker to determine MSI, was proposed by Kang et al. (2022) [27] due to its high sensitivity (94%) and specificity (98%) for determining MSI. The authors assessed MSI status in a series of cancers using multiplex PCR, which included markers BAT-25, BAT-26, NR-21, NR-24, and NR-27. They found that only 0.03% (2/6476) of the cases showed MSI-L positivity for the BAT-26 unique unstable marker. These cases were identified in patients with endometrial adenocarcinoma and colorectal carcinoma. This suggestion is relevant to us due to the high percentage of MSI-L PCa patients who showed BAT-26 instability (60.87% (14/23)), contrary to the lower positive cases (0.03% (2/6476)) that they reported.

Collectively, our investigation represents the first to evaluate MSI frequency in PCa among the Mexican population, and the first to make a distinction between MSI-H and MSI-L among this population. In addition, our results showed that the MSI-positive PCa patients exhibited a predominant Gleason score of 9—associated with a high aggressiveness—and a majority instability of the BAT-26 marker.

While our study offers valuable information, we were limited by the number of patients included in each Gleason score group and, therefore, the representativeness could be biased. Another limitation of this study is the lack of a database with the normal base pair length ranges of each marker, used to determine microsatellite instability in the Mexican population more specifically and sensitively.

It is essential to establish standardized criteria, ranges, and techniques for determining microsatellite instability (MSI) in prostate cancer (PCa) patients. The frequency of MSI in PCa has not been studied as extensively as in other cancers, such as endometrial, gastric, or colorectal cancer. Additionally, most reported MSI frequencies do not represent the Latin or Mexican populations, making it difficult for the medical and research community to develop more effective therapies tailored to Mexican patients. This study opens the door for future research in cancer and personalized therapy. Further studies are needed in different populations to advance personalized treatment for PCa or other types of cancers with MSI-positive cases, considering the promising therapies available for patients with these features.

## 5. Conclusions

This study represents the first effort to assess the prevalence of MSI in PCa among the Mexican population. Within the group of Mexican patients with MSI-positive PCa, Gleason score 9 was predominant. No significant differences were found between Gleason score groups and MSI-positive cases, nor between Gleason score groups and MSI-H and MSI-L. However, a weak correlation was suggested in both comparisons.

Additionally, there was a significant instability noted in the BAT-26 marker. Our results highlight the necessity of conducting population-focused assessments, as genetic and environmental factors may influence both the frequency and significance of MSI.

## Figures and Tables

**Figure 1 diseases-13-00202-f001:**
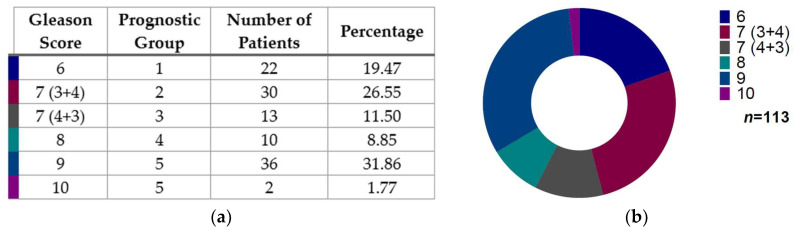
Gleason score distribution in prostate cancer patients: (**a**) Data distribution of prostate cancer patients indicated according to Gleason score and prognostic group. (**b**) Chart that shows patients’ distribution according to the Gleason score (*n* = 113). Note: The total study sample illustrates 116 patients, from which we had data from just 113 patients according to the Gleason score.

**Figure 2 diseases-13-00202-f002:**
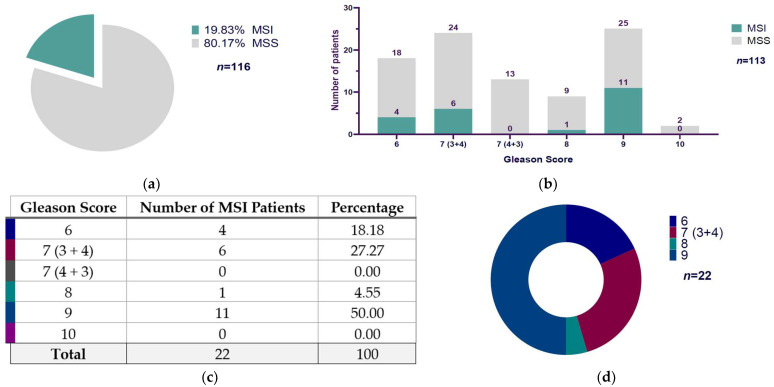
Presence of microsatellite instability and Gleason score distribution in prostate cancer patients: (**a**) Graph that shows the microsatellite instability (MSI) and microsatellite stable (MSS) percentage cases (MSI: 19.83% (23/116); MSS: 80.17% (93/116)) in prostate cancer patients regarding the total study sample (*n* = 116). (**b**) Bar chart that shows the number of MSI and MSS prostate cancer patients according to the Gleason score (*n* = 116). (**c**,**d**) Gleason score distribution related to total MSI prostate cancer patients (Gleason score 6: 18.18% (4/22); Gleason score 7 (3 + 4): 27.27% (6/22); Gleason score 7 (4 + 3): 0% (0/22); Gleason score 8: 4.55% (1/22); Gleason score 9: 50% (11/22); Gleason score 10: 0% (0/22); *n* = 22). Note: The total number of patients indicated in (**b**) (*n* = 113) is different from the number indicated in (**a**) (*n* = 116) because we had no Gleason score information for 3 patients (1 MSI and 2 MSS). MSS: microsatellite stable; MSI: microsatellite instability.

**Figure 3 diseases-13-00202-f003:**
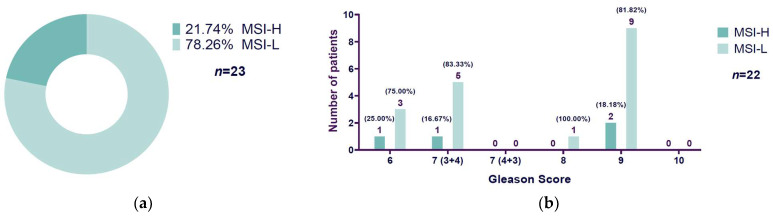
Microsatellite instability-high and -low of prostate cancer patients: (**a**) The chart shows microsatellite instability-high (MSI-H) and microsatellite instability-low (MSI-L) distribution of prostate cancer patients (MSI-H: 21.74% (5/23); MSI-L: 78.26% (18/23); *n* = 23). (**b**) Bar chart that shows MSI-H and MSI-L prostate cancer patients’ distribution in relation to their Gleason score (*n* = 22). It is specified between parentheses the percentage of patients in relation to the MSI-H and MSI-L total patients, respectively, per each Gleason score. Note: The sample size in (**a**) (*n* = 23) is different to (**b**) (*n* = 22) because we had no Gleason score information for 1 MSI-H patient. MSI-L: microsatellite instability-low; MSI-H: microsatellite instability-high.

**Figure 4 diseases-13-00202-f004:**
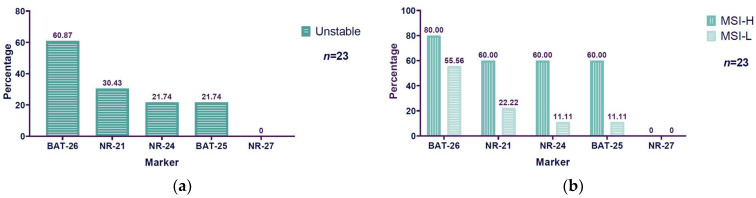
Unstable markers in microsatellite instability prostate cancer patients: (**a**) Bar chart shows unstable marker percentage in MSI prostate cancer patients (BAT-26: 60.87% (14/23); NR-21: 30.43% (7/23); NR-24: 21.74% (5/23); BAT-25: 21.74% (5/23); NR-27: 0% (0/23); *n* = 23). (**b**) The bar chart shows a distinction between the unstable marker percentage presented in MSI-H patients (BAT-26: 80% (4/5); NR-21: 60% (3/5); NR-24: 60% (3/5); BAT-25: 60% (3/5); NR-27: 0% (0/5)) and in MSI-L patients (BAT-26: 55.56% (10/18); NR-21: 22.22% (4/18); NR-24: 11.11% (2/18); BAT-25: 11.11% (2/18); NR-27: 0% (0/18); *n* = 23). Note: Markers were ordered according to the instability percentage illustrated in the study. MSI: microsatellite instability; MSI-L: microsatellite instability-low; MSI-H: microsatellite instability-high.

**Table 1 diseases-13-00202-t001:** Contingency table of Gleason scores and microsatellite instability status.

Gleason Score	MSI	MSS	Total
6	4	18	22
7 (3 + 4)	6	24	30
7 (4 + 3)	0	13	13
8	1	9	10
9	11	25	36
10	0	2	2

**Table 2 diseases-13-00202-t002:** Contingency table of Gleason scores and microsatellite instability-high and microsatellite instability-low status.

Gleason Score	MSI-L	MSI-H	Total
6	1	3	4
7 (3 + 4)	1	5	6
7 (4 + 3)	0	0	0
8	0	1	1
9	2	9	11
10	0	0	0

**Table 3 diseases-13-00202-t003:** Contingency table of microsatellite instability-high and microsatellite instability-low status and the BAT-26 marker.

BAT-26	MSI-L	MSI-H	Total
Stable	8	1	9
Unstable	10	4	14

## Data Availability

The data that support the findings of this study are available from the corresponding authors (adrian.ramirez@academicos.udg.mx and miguel.moreno@academicos.udg.mx) upon reasonable request.

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
