# Peer review of "Microsatellite Instability and BAT-26 Marker Expression in a Mexican Prostate Cancer Population with Different Gleason Scores"

_diseases, 2025, doi:10.3390/diseases13070202_

Round 1
Reviewer 1 Report
Comments and Suggestions for Authors
Ana K. Flores-Islas et al presented an interested article. The authors explores and evaluate the frequency of MSI in 113 Mexican PCa patients and investigated its potential relationship with tumor aggressiveness. MSI was performed by multiplex PCR with a panel of 5 markers (NR- 53
27, NR-21, NR-24, BAT-25, and BAT-26).
Major concerns,
While the multiplex panel is widely acceptable in clinical and research. It was more reliable if the authors perform next generation seq to avoid false positive or other PCR related issues.
Also sample size is too small.
minor comments
Methods section, please add patient samples demographics.
Statistics and Analysis of data section not available in the MM section.
The data in the result section presented in percentage only.
Author Response
Ana K. Flores-Islas et al presented an interested article. The authors explores and evaluate the frequency of MSI in 113 Mexican PCa patients and investigated its potential relationship with tumor aggressiveness. MSI was performed by multiplex PCR with a panel of 5 markers (NR-27, NR-21, NR-24, BAT-25, and BAT-26).
Major concerns,
While the multiplex panel is widely acceptable in clinical and research. It was more reliable if the authors perform next generation seq to avoid false positive or other PCR related issues.
Answer: We appreciate the reviewer’s comment. The pentaplex MSI panel is a simple, PCR- and capillary electrophoresis-based technique that offers the advantage of being easily implemented in most laboratories without the need for expensive equipment or highly specialized personnel. It also provides results in a significantly shorter turnaround time. This panel has demonstrated high sensitivity (95.6%) and a positive predictive value of 100% for MSI detection, making it a reliable method for classifying tumors as MSI-H, MSI-L, or MSS. Importantly, it does not require matched normal DNA for interpretation and has a low per-sample cost compared to next-generation sequencing (NGS), which makes it a viable option for population-based studies. Lastly, this panel is FDA-approved for guiding therapeutic decisions, supporting its clinical utility in the context of personalized cancer treatment.
Also sample size is too small.
Answer: We appreciate the reviewer’s insightful observation. The current study was based on a retrospective analysis of prostate cancer tissue samples obtained from a histopathological tissue bank. As such, our access was limited to the archived biological material, and unfortunately, no accompanying demographic or clinical data (e.g., age, comorbidities, treatment history) were available. This also constrained the total number of samples included (n = 116), as only cases with adequate preserved tissue for analysis were selected. We acknowledge this limitation and have clarified it in the revised manuscript.
Besides, based on two representative studies showing the prevalence of MSI in prostate cancer (Abida 2019, DOI: 10.1001/jamaoncol.2018.5801; Chung 2019 DOI: https://doi.org/10.1200/PO.18.00283) that reported 3.1% and 2.6% respectively we performed the sample size calculation using the OpenEpi website. In order to achieve 99.9% confidence interval, 110 samples are required.
minor comments
Methods section, please add patient samples demographics.
Answer: We appreciate the reviewer’s insightful observation. The current study was based on a retrospective analysis of prostate cancer tissue samples obtained from a histopathological tissue bank. As such, our access was limited to the archived biological material, and unfortunately, no accompanying demographic or clinical data (e.g., age, comorbidities, treatment history) were available. This also constrained the total number of samples included (n = 116), as only cases with adequate preserved tissue for analysis were selected. We acknowledge this limitation and have clarified it in the revised manuscript.
Statistics and Analysis of data section not available in the MM section.
Answer: Statistical analysis was added in Materials and Methods section (lines 164-171)
The data in the result section presented in percentage only.
Answer: Contingency tables were added for each result to clarify the analysis, where the number of cases is presented instead of the percentage (lines 217, 247, 286).
Finally, in consideration of the reviewers’ valuable suggestions and after implementing the corresponding revisions, we believe the title of the manuscript would be more appropriately presented as:
“Microsatellite Instability and BAT26 Marker Expression in a Mexican Prostate Cancer Population with Different Gleason Scores.”
Reviewer 2 Report
Comments and Suggestions for Authors
The authors of the study investigated the relationship between microsatellite instability (MSI) and histological grade using Gleason score on prostate cancers. The research revealed that MSI-positive samples were uncommon and primarily identified in Gleason score 9, which is histological high grade, in prostate cancer.
The limited number of cases available for analysis has led to the generation of ambiguous results, which is a shortcoming. The only 116 cases were used in the study, and the low positivity rate of the MSI cases indicated that the limited number of MSI cases available for analysis precluded a detailed study. Additionally, the paucity of clinicopathological data from 116 prostate cancer cases imposes constraints on the analysis of MSI.
Another problem is the difference of analysis cases in some experiments. A total of 113 cases were utilized for the histological analysis (Gleason). However, 116 cases were used for some part of the MSI analysis, which caused confusion in the resulting numbers of cases. They should use 113 cases in this study.
Author Response
The authors of the study investigated the relationship between microsatellite instability (MSI) and histological grade using Gleason score on prostate cancers. The research revealed that MSI-positive samples were uncommon and primarily identified in Gleason score 9, which is histological high grade, in prostate cancer.
The limited number of cases available for analysis has led to the generation of ambiguous results, which is a shortcoming. The only 116 cases were used in the study, and the low positivity rate of the MSI cases indicated that the limited number of MSI cases available for analysis precluded a detailed study. Additionally, the paucity of clinicopathological data from 116 prostate cancer cases imposes constraints on the analysis of MSI.
Answer: We appreciate the reviewer’s insightful observation. The current study was based on a retrospective analysis of prostate cancer tissue samples obtained from a histopathological tissue bank. As such, our access was limited to the archived biological material, and unfortunately, no accompanying demographic or clinical data (e.g., age, comorbidities, treatment history) were available. This also constrained the total number of samples included (n = 116), as only cases with adequate preserved tissue for analysis were selected. We acknowledge this limitation and have clarified it in the revised manuscript.
Besides, based on two representative studies showing the prevalence of MSI in prostate cancer (Abida 2019, DOI: 10.1001/jamaoncol.2018.5801; Chung 2019 DOI: https://doi.org/10.1200/PO.18.00283) that reported 3.1% and 2.6% respectively we performed the sample size calculation using the OpenEpi website. In order to achieve 99.9% confidence interval, 110 samples are required.
Another problem is the difference of analysis cases in some experiments. A total of 113 cases were utilized for the histological analysis (Gleason). However, 116 cases were used for some part of the MSI analysis, which caused confusion in the resulting numbers of cases. They should use 113 cases in this study.
Answer: Thank you to the reviewer for this observation. We had a total of 116 patient samples available for MSI determination; for that reason, although we didn’t have Gleason Score information for all 116 patients (we could only obtain Gleason score data for 113), we decided to perform MSI determination for the valuable information we would gain. However, to avoid any confusion, we added extra explanations for each Result that mentions different sample sizes, which can cause confusion (lines 188-189, 221-222).
Finally, in consideration of the reviewers’ valuable suggestions and after implementing the corresponding revisions, we believe the title of the manuscript would be more appropriately presented as:
“Microsatellite Instability and BAT26 Marker Expression in a Mexican Prostate Cancer Population with Different Gleason Scores.”
Reviewer 3 Report
Comments and Suggestions for Authors
The authors evaluated the microsatellite instability (MSI) in 116 Mexican prostate cancer patients and explored the relationship between MSI and Gleason score.
Some comments are listed below.
- In the introduction, please include how microsatellite instability is determined in prostate and other cancers and which methods or biomarkers are used.
- The authors should list all the clinical and pathological characteristics and genetic features of 116 patients, including age, PSA, clinical T stage, pathological stage, Gleason score, metastasis, and biochemical recurrence.
- Using microsatellite instability-high (MSI-H), microsatellite instability-low (MSI-L), and microsatellite stable (MSS) to categorize 116 patients and evaluate the differences in clinical and pathological characteristics and survival by statistics.
- Figures 1-4 should indicate if there are significant differences or not between groups.
- Lines 289-291: “The findings are significant, particularly given that only patients with high microsatellite instability (MSI-H) are considered candidates for immunotherapy, while those with low instability (MSI-L) are not.” Please elaborate on why MSI-H are considered candidates for immunotherapy in more detail. Are there supporting studies on prostate cancer?
- Line 353: “there was a significant instability noted in the BAT-353 26 marker.”, please demonstrate the significant difference by statistics.
Author Response
The authors evaluated the microsatellite instability (MSI) in 116 Mexican prostate cancer patients and explored the relationship between MSI and Gleason score.
Some comments are listed below.
In the introduction, please include how microsatellite instability is determined in prostate and other cancers and which methods or biomarkers are used.
Answer: We appreciate the reviewer’s insightful observation. We have briefly added the most common techniques used to determine MS in the Introduction section (lines 90-97).
The authors should list all the clinical and pathological characteristics and genetic features of 116 patients, including age, PSA, clinical T stage, pathological stage, Gleason score, metastasis, and biochemical recurrence.
Answer: We appreciate the reviewer’s insightful observation. The current study was based on a retrospective analysis of prostate cancer tissue samples obtained from a histopathological tissue bank. As such, our access was limited to the archived biological material, and unfortunately, no accompanying demographic or clinical data (e.g., age, PSA, clinical T stage, pathological stage, metastasis, and biochemical recurrence) were available. This also constrained the total number of samples included (n = 116), as only cases with adequate preserved tissue for analysis were selected. We acknowledge this limitation and have clarified it in the revised manuscript.
Using microsatellite instability-high (MSI-H), microsatellite instability-low (MSI-L), and microsatellite stable (MSS) to categorize 116 patients and evaluate the differences in clinical and pathological characteristics and survival by statistics.
Answer: Statistical analysis were performed and even though no significant differences were found, the result of this analysis is shown and discussed in the article. Contingency tables were added for each result to clarify the analysis. Unfortunately, following the answer in point number 2, our access was limited to the archived biological material, and unfortunately, no accompanying demographic or clinical data (e.g., age, PSA, clinical T stage, pathological stage, metastasis, and biochemical recurrence) were available, due to the prostate cancer tissue samples were obtained from a histopathological tissue bank. However, these analyses are planned as part of a follow-up study using a different cohort, specifically designed for extended molecular profiling.
Figures 1-4 should indicate if there are significant differences or not between groups.
Answer: Statistical analyses were performed and even though no significant differences were found, the result of this analysis is shown and discussed in the article. Contingency tables were added for each result to clarify the analysis (lines 209-217, 238-248, 275-287).
Lines 289-291: “The findings are significant, particularly given that only patients with high microsatellite instability (MSI-H) are considered candidates for immunotherapy, while those with low instability (MSI-L) are not.” Please elaborate on why MSI-H are considered candidates for immunotherapy in more detail. Are there supporting studies on prostate cancer?
Answer: The recommendations for immunotherapy use in prostate cancer tumors with MSI-H (specifically MSI-H, not MSI-L); dMMR; TMB-high were cited (lines 349-353).
Line 353: “there was a significant instability noted in the BAT-353 26 marker.”, please demonstrate the significant difference by statistics.
Answer: Statistical analyses were performed and the results were added in the text and the table 3 (lines 275-287).
Finally, in consideration of the reviewers’ valuable suggestions and after implementing the corresponding revisions, we believe the title of the manuscript would be more appropriately presented as:
“Microsatellite Instability and BAT26 Marker Expression in a Mexican Prostate Cancer Population with Different Gleason Scores.”
Reviewer 4 Report
Comments and Suggestions for Authors
The paper is a retrospective analysis of a series of patients with PCA. Authors assessed the incidence of MSI among PCA patients from formalin-fixed paraffin-embedded tumoral tissue and fresh tumoral tissue. Authors found an unusually high incidence of MSI that jumps to 50% in ISUP 5 cases. Their paper is interesting inasmuch, it confirms that ISUP 4 and 5 cancers may be considered different diseases biologically and clinically
1) Authors state that formalin-fixed paraffin-embedded tumoral tissue and fresh tumoral tissue was analyzed but, when it comes to DNA extraction, they refer only to formalin-fixed paraffin-embedded tumoral tissue. Was fresh tissue analyzed? And how?
2) The series is composed of 113 cases. How did the Authors calculate the sample size?
3) How do Authors explain the 50% incidence in Gleason 9 and 0% in Gleason 10 patients
Author Response
The paper is a retrospective analysis of a series of patients with PCA. Authors assessed the incidence of MSI among PCA patients from formalin-fixed paraffin-embedded tumoral tissue and fresh tumoral tissue. Authors found an unusually high incidence of MSI that jumps to 50% in ISUP 5 cases. Their paper is interesting inasmuch, it confirms that ISUP 4 and 5 cancers may be considered different diseases biologically and clinically
1) Authors state that formalin-fixed paraffin-embedded tumoral tissue and fresh tumoral tissue was analyzed but, when it comes to DNA extraction, they refer only to formalin-fixed paraffin-embedded tumoral tissue. Was fresh tissue analyzed? And how?
Answer: We appreciate the reviewer’s helpful observation. It’s correct, we did not use fresh tumoral tissue, for that reason, that sentence was eliminated.
2) The series is composed of 113 cases. How did the Authors calculate the sample size?
Answer: Based on two representative studies showing the prevalence of MSI in prostate cancer (Abida 2019 DOI: 10.1001/jamaoncol.2018.5801, Chung 2019 DOI: https://doi.org/10.1200/PO.18.00283) that reported 3.1% and 2.6% respectively we performed the sample size calculation using the OpenEpi website. In order to achieve 99.9% confidence interval, 110 samples are required.
3) How do Authors explain the 50% incidence in Gleason 9 and 0% in Gleason 10 patients
Answer: According to ISUP 2019, the prognostic grade five is composed by Gleason 9 and 10, which means that there are equivalent in the clinical value and therefore they can be contemplated as the same group.
Finally, in consideration of the reviewers’ valuable suggestions and after implementing the corresponding revisions, we believe the title of the manuscript would be more appropriately presented as:
“Microsatellite Instability and BAT26 Marker Expression in a Mexican Prostate Cancer Population with Different Gleason Scores.”
Reviewer 5 Report
Comments and Suggestions for Authors
The title accurately reflects the content of the manuscript. The simple summary and abstract are reasonably well written. However, the sentence describing the MSI-L has to be more clearly written: ...low microsatellite instability (MSI-L) as an unstable marker....do you mean that MSI was present on one marker only?
My concern is the sample number, which is rather low for such a common cancer. Moreover, you claim that 23/116 were MSI PCa patients, of which 21.74% (5/23) were MSI-H, and 78.26% (18/23) 57 were MSI-L. Since MSI at just one marker can be considered an accidental finding, and some studies consider the positive samples on just one marker as stable. Based on this, you only have 5 out of 116 MSI high samples, which might be considered suitable for immunotherapy. The panel you used is not the standard Bethesda panel, which is used in most studies therefore your findings are not easily compared to others. I would suggest that you include some markers from the Bethesda panel and/or IHC for MMR proteins and reanalyze your samples.
Finally the number of authors seems rather high, and from the authors' contribution paragraph, it seems that all they did was write, supervise, project manage etc. -it is not clear who did the work with their own hands.
Therefore since there are as many authors I would suggest that you add some additional markers to your analysis -at least to these MSI-L samples.
Moreso, the discussion is quite long, but there is no mention of the new emerging concept of instability at longer microsatellite repeats which might be the intermediate form of microsatellite instability between MSI high and MSS as for instance described in https://pubmed.ncbi.nlm.nih.gov/37510378/
Author Response
The title accurately reflects the content of the manuscript. The simple summary and abstract are reasonably well written. However, the sentence describing the MSI-L has to be more clearly written: ...low microsatellite instability (MSI-L) as an unstable marker....do you mean that MSI was present on one marker only?
Answer: We explained the interpretation of microsatellite instability in the Materials and Methods section (lines 156-163). However, we rewrote that sentence to avoid any doubt Additionally, the different techniques used to determine MSI were briefly described in the Introduction section (lines 90-97)
My concern is the sample number, which is rather low for such a common cancer.
Answer: We appreciate the reviewer’s insightful observation. Based on two representative studies showing the prevalence of MSI in prostate cancer (Abida 2019 DOI: 10.1001/jamaoncol.2018.5801, Chung 2019 DOI: https://doi.org/10.1200/PO.18.00283) that reported 3.1% and 2.6% respectively we performed the sample size calculation using the OpenEpi website. In order to achieve 99.9% confidence interval, 110 samples are required.
Besides, the current study was based on a retrospective analysis of prostate cancer tissue samples obtained from a histopathological tissue bank and only cases with adequate preserved tissue for analysis were selected, which might have constrained the total number of samples included (n = 116). We acknowledge this limitation and have clarified it in the revised manuscript.
Moreover, you claim that 23/116 were MSI PCa patients, of which 21.74% (5/23) were MSI-H, and 78.26% (18/23) 57 were MSI-L. Since MSI at just one marker can be considered an accidental finding, and some studies consider the positive samples on just one marker as stable. Based on this, you only have 5 out of 116 MSI high samples, which might be considered suitable for immunotherapy.
Answer: Although the size of MSI-L samples may appear small, Murphy et al. (2006) note that the presence of a single unstable marker would be highly suspicious and would warrant further testing, such as monitoring patients with MSI-L, since tumor progression could lead to additional markers becoming unstable and changing classification to MSI-H
The panel you used is not the standard Bethesda panel, which is used in most studies therefore your findings are not easily compared to others. I would suggest that you include some markers from the Bethesda panel and/or IHC for MMR proteins and reanalyze your samples.
Answer: We thank the reviewer for this observation. The panel used in our study was selected over the original Bethesda panel (1997) because it provides higher analytical performance, with a reported sensitivity of 95.6% and a positive predictive value of 100% for the detection of microsatellite instability (Goel et al., 2010). Unlike the Bethesda panel, our assay includes five quasi-monomorphic mononucleotide markers, whose allelic sizes are virtually identical across global populations. Notably, two markers from the Bethesda panel are also included in the pentaplex panel we employed. Additionally, our panel is FDA-approved for guiding immunotherapy decisions. In contrast, the Bethesda panel presents certain limitations, such as the inclusion of dinucleotide markers and the requirement for matched normal tissue for comparison—an approach that can be particularly challenging in cancers like prostate cancer.
Finally the number of authors seems rather high, and from the authors' contribution paragraph, it seems that all they did was write, supervise, project manage etc. -it is not clear who did the work with their own hands.
Answer: The author's paragraph has been written in accordance with the journal’s Word template. However, I would like to clarify that the person who did the work with their own hands, indicated as “writing-original draft preparation”, was Ana Karen Flores Islas, identified as A.K.F.-I. (lines 440-447)
Therefore since there are as many authors I would suggest that you add some additional markers to your analysis -at least to these MSI-L samples.
Answer: We appreciate the reviewer’s thoughtful suggestion. The main objective of the current study was to identify the presence of microsatellite instability (MSI) in prostate cancer samples from Mexican patients using the available tissue bank. We fully agree that incorporating additional molecular markers, such as methylation status, would provide deeper insights. However, these analyses are planned as part of a follow-up study using a different cohort, specifically designed for extended molecular profiling. We have clarified this point in the Discussion section to better reflect the scope and limitations of the present work.
Moreso, the discussion is quite long, but there is no mention of the new emerging concept of instability at longer microsatellite repeats which might be the intermediate form of microsatellite instability between MSI high and MSS as for instance described in https://pubmed.ncbi.nlm.nih.gov/37510378/
Answer: We appreciate the valuable contribution. An extra paragraph was incorporated in the discussion section with that information (lines 357-374)
Finally, in consideration of the reviewers’ valuable suggestions and after implementing the corresponding revisions, we believe the title of the manuscript would be more appropriately presented as:
“Microsatellite Instability and BAT26 Marker Expression in a Mexican Prostate Cancer Population with Different Gleason Scores.”
Round 2
Reviewer 3 Report
Comments and Suggestions for Authors
I agree to accept the manuscript.
Reviewer 5 Report
Comments and Suggestions for Authors
nothing further